

# COVID-19 mortality in cancer patients: a report from a tertiary cancer centre in India

Anurag Mehta[1], Smreti Vasudevan[2], Anuj Parkash[3], Anurag Sharma[2], Tanu Vashist[2] and Vidya Krishna[2]

[1] Department of Laboratory, Transfusion and Molecular Diagnostics Services, Rajiv Gandhi Cancer Institute & Research Centre, New Delhi, Delhi, India
[2] Department of Research, Rajiv Gandhi Cancer Institute & Research Centre, New Delhi, Delhi, India
[3] Department of Laboratory and Transfusion Services, Rajiv Gandhi Cancer Institute & Research Centre, New Delhi, Delhi, India

Corresponding authors
Smreti Vasudevan,
smreti@gmail.com
Anuj Parkash, dranujpa@gmail.com

## ABSTRACT

**Background:** Cancer patients, especially those receiving cytotoxic therapy, are assumed to have a higher probability of death from COVID-19. We have conducted this study to identify the Case Fatality Rate (CFR) in cancer patients with COVID-19 and have explored the relationship of various clinical factors to mortality in our patient cohort.

**Methods:** All confirmed cancer cases presented to the hospital from June 8 to August 20, 2020, and developed symptoms/radiological features suspicious of COVID-19 were tested by Real-time polymerase chain reaction assay and/or cartridge-based nucleic acid amplification test from a combination of naso-oropharyngeal swab for SARS-CoV-2. Clinical data, treatment details, and outcomes were assessed from the medical records.

**Results:** Of the total 3,101 cancer patients admitted to the hospital, 1,088 patients were tested and 186 patients were positive for SARS-CoV-2. The CFR in the cohort was 27/186 (14.52%). Univariate analysis showed that the risk of death was significantly associated with the presence of any comorbidity (OR: 2.68; (95% CI [1.13–6.32]); $P = 0.025$), multiple comorbidities (OR: 3.01; (95% CI [1.02–9.07]); $P = 0.047$ for multiple vs. single), and the severity of COVID-19 presentation (OR: 27.48; (95% CI [5.34–141.49]); $P < 0.001$ for severe vs. not severe symptoms). Among all comorbidities, diabetes (OR: 3.31; (95% CI [1.35–8.09]); $P = 0.009$) and cardiovascular diseases (OR: 3.77; (95% CI [1.02–13.91]); $P = 0.046$) were significant risk factors for death. Anticancer treatments including chemotherapy, surgery, radiotherapy, targeted therapy, and immunotherapy administered within a month before the onset of COVID-19 symptoms had no significant effect on mortality.

**Conclusion:** To the best of our knowledge, this is the first study from India reporting the CFR, clinical associations, and risk factors for mortality in SARS-CoV-2 infected cancer patients. Our study shows that the frequency of COVID-19 in cancer patients is high. Recent anticancer therapies are not associated with mortality. Pre-existing comorbidities, especially diabetes, multiple

comorbidities, and severe symptoms at presentation are significantly linked with COVID-19 related death in the cohort.

# INTRODUCTION

The ongoing pandemic of coronavirus disease 2019 (COVID-19) caused by severe acute respiratory syndrome coronavirus 2 (SARS-CoV-2) has caused unprecedented health and societal crises across the globe. However, this catastrophe has victimized the cancer patients the most, adversely impacting diagnosis and treatment in about 55% of cases worldwide (*World Health Organisation, 2020*). The number of patients visiting and accessing oncological services has considerably reduced and the collateral damage can have an adverse impact on cancer outcomes (*Guven et al., 2020*; *Sud et al., 2020*). It is generally assumed that patients with cancer are at a higher risk of contracting COVID-19 due to their immunosuppressive state, and side effects associated with anticancer therapies like leukopenia and disruption of the barrier to infections (*ElGohary et al., 2020*; *Ganatra, Hammond & Nohria, 2020*; *Liang et al., 2020*). Additionally, it is also postulated that the severity and resulting mortality is amplified in cancer patients with COVID-19 due to their elderly and the immunocompromised state further worsened by cancer treatment. Multiple studies from different geographical locations show that the case fatality rate (CFR) of the SARS-CoV-2 infected cancer patients varies from 3.7% to 61.5% (*He et al., 2020*; *Kalinsky et al., 2020*; *Saini et al., 2020*). Despite recent efforts, no clear consensus has been reached for the relation of mortality to demographics, cancer type, stage, and underlying comorbidities; and it may depend on the epidemiology and prevalent oncology practices (*Miyashita et al., 2020*; *Oh, 2020*). Various studies regarding the risk of treating cancer patients during the pandemic have shown contradictory results (*Dai et al., 2020*; *Lee et al., 2020*; *Liang et al., 2020*; *Pinato et al., 2020*). Larger datasets acquired globally from different sources and geographic locations and rationally analyzed has the best potential to provide an effective risk-benefit calculus to assist oncologists to optimize the management and use anticancer treatments to restore outcomes to the pre-COVID-19 era.

We aim to describe the clinical and demographic characteristics and COVID-19 outcomes in a cohort of patients with cancer and symptomatic COVID-19. An attempt has also been made to assess the adverse effect of cytotoxic and other novel therapies on mortality in cancer patients with COVID-19.

# MATERIALS AND METHODS

## Study design and subjects

The study is single-center, retrospective, conducted at a tertiary cancer care hospital. Patients with active cancer presented to the hospital between June 8 and August 20, 2020, and with confirmed COVID-19 infection were included. Clinical data, anticancer

treatment details (performed within a month of COVID-19 diagnosis), clinical course, and outcome were retrieved from the hospital electronic medical records. The COVID-19 infection severity of patients was scored at the time of presentation by the treating physician, according to the Ministry of Health and Family Welfare (Government of India) guidelines (*Government of India Ministry of Health and Family Welfare, 2020a*, *2020b*).

The study has been approved by our Institutional Review Board (RGCIRC/IRB-BHR/61/2020) and was conducted according to the Declaration of Helsinki.

### Real-time polymerase chain reaction assay

According to the Indian Council of Medical Research (ICMR) guidelines and international practice, the COVID-19 symptomatic cancer patients were tested by Real-Time Polymerase Chain Reaction (RT-PCR) assay and/or cartridge-based nucleic acid amplification test (CBNAAT) for SARS-CoV-2.

Samples were collected from the nasopharynx and oropharynx in a single tube with 3 ml of the viral transport medium (Biogenix®, Uttar Pradesh, India). The RT-PCR test was carried out on the QuantStudio™ 5 Real-Time PCR System (Thermo-Fishers Scientific-Life Technologies Holdings Pvt. Ltd., Singapore) employing an ICMR approved RT-PCR test kit (TRUPCR®-3B; BlackBio Biotech, Madhya Pradesh, India). This kit detects the E gene common to the Sarbecovirus superfamily, a sampling control of human ribonucleic acid nuclease P (hRNaseP) and RNA dependent RNA polymerase (RdRP) gene, and N gene for the detection of SARS-CoV-2. The test was performed and interpreted as per the manufacturer's instructions.

In certain cases, rapid molecular testing was used utilizing CBNAAT from Cepheid® (Sunnyvale, CA, USA) (GeneXpert® assay). It identifies E gene-specific RNA and N2 gene-specific RNA. The test was performed and interpreted as per the downloaded package insert.

### Statistical analysis

Continuous variables were presented as mean±standard deviation (SD) or median (interquartile range, IQR). Categorical variables were presented as frequencies and percentages. The two-sided independent *t*-test and the median test were used to compare the mean and median ages, respectively. Fisher's exact test/Pearson's Chi-Squared test was used to compare categorical data. Univariate logistic regression was used to estimate odds ratio (OR) and 95% confidence intervals (CI). Multivariate logistic regression was used to compute the odds ratio for the various treatment modalities after adjusting for age and comorbidities. All the statistical analyses have been performed either by using SPSS® Version 23.0 software or MedCalc Statistical Software version 19.4.0. The reported *P* values are two-sided and a *P* value < 0.05 was considered statistically significant.

## RESULTS

A total of 3,101 cancer patients were treated at the indoor facility of the center. One thousand and eighty-eight patients developed signs, symptoms, and/or radiological features suspicious of COVID-19. Of these, 186 tested positive for COVID-19 and formed

the study cohort. The infection rate of COVID-19 among all cancer patients treated at the center was ~6% (186/3,101) and 17.1% (186/1,088) of the symptomatic and tested cancer patients. The clinical features are shown in Table 1. Most patients had solid malignancies (82.26%); gastrointestinal cancer (21.51%) was the most common cancer type, and about 17.74% of cases presented with hematological malignancies. More than a quarter of cases (26.88%) were metastatic. Eighty-six patients (46.24%) had at least a single comorbidity; hypertension (24.19%) and diabetes (18.28%) were the most common. About 60% of cases were on active cancer treatment and had received cancer-directed treatment within a month before the onset of COVID-19 symptoms. The majority of patients were on chemotherapy (37%).

The chief presenting symptoms of SARS-CoV-2 infection were fever (123/186, 66.13%), fatigue (26/186, 13.98%) and respiratory distress (25/186, 13.44%). Around 5% of patients were presented with severe disease. The COVID-19 associated fatality rate in the cohort was 14.52% (27/186) (median follow-up duration: 63 days). Compared to hospitalized cancer patients without COVID-19, the CFR in cancer patients with COVID-19 was significantly higher (27/186, 14.32% vs. 40/2,915, 1.37%; $P < 0.0001$). The CFR for hematological malignancies tended to be higher than the CFR for solid malignancies, however, the difference did not reach statistical significance (7/33, 21.21% vs. 20/153, 13.07%, $P = 0.274$).

Severe COVID-19 infected cancer cases were managed by treatments including corticosteroids, hydroxychloroquine, remdesivir, tocilizumab and convalescent plasma therapy. Patients with mild or moderate disease severity were given symptomatic treatment. Hydroxychloroquine and/or dexamethasone were administered in moderate disease severity cases. Assisted ventilation was given to 12 patients (6.45%), however, all of these patients eventually developed COVID-19 related complications like pneumonitis and associated respiratory failure, septic shock, or sudden cardiac arrest and succumbed to the disease.

Next, we explored the differences between the cancer patients who died and those who survived the SARS-CoV-2 infection (Table 1). There was no significant difference between the survivors and the non-survivors with respect to age, gender, type of malignancy, and cancer spread. Also, no significant effect on mortality was noted for the patients who had received anticancer therapy within the past month. Deceased patients displayed significantly higher rates of comorbidity compared to the cancer patients who survived (66.67% vs. 42.77%, $P = 0.035$); patients with greater than one comorbidity had significantly inferior outcome than those with single or no comorbidity ($P = 0.004$). Importantly, patients with diabetes experienced significantly more deaths than patients without diabetes ($P = 0.013$) (Table 1). Further, we observed that cancer patients presented with moderate or severe COVID-19 symptoms had significantly higher mortality than those presented with mild symptoms ($P < 0.00001$).

The univariate logistic regression analysis for death has been shown in Table 2. The mortality risk was statistically significant for the presence of any comorbidity (OR = 2.68, $P = 0.025$), multiple vs. single morbidity (OR = 3.01, $P = 0.047$), cardiovascular disease (OR = 3.77, $P = 0.046$) and diabetes (OR = 3.31, $P = 0.009$). The odds of death were

**Table 1 Clinical characteristics of cancer patients infected with COVID-19 according to outcome (N = 186).**

| | Total N = 186 (%) | Survivors n = 159 (%) | Non-survivors n = 27 (%) | P value |
|---|---|---|---|---|
| Age (years) | | | | |
| Mean ± SD | 50.24 ± 15.77 | 50.22 ± 15.66 | 50.33 ± 16.75 | 0.974 |
| Median (IQR) | 52 (42–58.75) | 52 (42–58.5) | 53 (43.5–60) | 0.952 |
| Gender | | | | |
| Male | 105 (56.45) | 89 (55.97) | 16 (59.26) | 0.835 |
| Female | 81 (43.55) | 70 (44.03) | 11 (40.74) | |
| Comorbidities | | | | |
| Present | 86 (46.24) | 68 (42.77) | 18 (66.67) | 0.035 |
| Absent | 100 (53.76) | 91 (57.23) | 9 (33.33) | |
| Number of comorbidities in a patient | | | | |
| No comorbidity | 100 (53.76) | 91 (57.23) | 9 (33.33) | 0.004 |
| Single | 47 (25.27) | 41 (25.79) | 6 (22.22) | |
| More than one | 39 (20.97) | 27 (16.98) | 12 (44.44) | |
| Type of comorbidities | | | | |
| Cardiovascular disease | 11 (5.91) | 7 (4.40) | 4 (14.81) | 0.057 |
| Chronic obstructive pulmonary disease | 2 (1.08) | 2 (1.26) | 0 (0.00) | 1.000 |
| Diabetes | 34 (18.28) | 24 (15.09) | 10 (37.04) | 0.013 |
| Hypertension | 45 (24.19) | 36 (22.64) | 9 (33.33) | 0.233 |
| Thyroid (Hypo/Hyper) | 30 (16.13) | 26 (16.35) | 4 (14.81) | 1.000 |
| Other comorbidities | 5 (2.69) | 3 (1.89) | 2 (7.41) | 0.154 |
| Solid vs. Hematological cancer | | | | |
| Solid | 153 (82.26) | 133 (83.65) | 20 (74.07) | 0.274 |
| Hematological | 33 (17.74) | 26 (16.35) | 7 (25.93) | |
| Cancer type | | | | |
| Brain | 2 (1.08) | 2 (1.26) | 0 (0.00) | 0.095 |
| Head and neck | 33 (17.74) | 27 (16.98) | 6 (22.22) | |
| Breast | 19 (10.22) | 18 (11.32) | 1 (3.70) | |
| Thoracic | 17 (9.14) | 17 (10.69) | 0 (0.00) | |
| Musculoskeletal and skin | 6 (3.23) | 4 (2.52) | 2 (7.41) | |
| Gastrointestinal | 40 (21.51) | 31 (19.50) | 9 (33.33) | |
| Genitourinary and gynecologic | 36 (19.35) | 34 (21.38) | 2 (7.41) | |
| Hematological | 33 (17.74) | 26 (16.35) | 7 (25.93) | |
| Cancer spread | | | | |
| Localized tumor | 58 (31.18) | 52 (32.70) | 6 (22.22) | 0.361 |
| Locally advanced | 78 (41.94) | 67 (42.14) | 11 (40.74) | |
| Metastatic | 50 (26.88) | 40 (25.16) | 10 (37.04) | |
| Cancer directed treatment within 1 month of COVID-19 infection | | | | |
| Treated cases | 112 (60.22) | 94 (59.12) | 18 (66.67) | 0.528 |
| No treatment | 74 (39.78) | 65 (40.88) | 9 (33.33) | |

(Continued)

|  | Total N = 186 (%) | Survivors n = 159 (%) | Non-survivors n = 27 (%) | P value |
|---|---|---|---|---|
| Cancer treatment |  |  |  |  |
| Surgery | 31 (16.67) | 27 (16.98) | 4 (14.81) | 1.000 |
| Chemotherapy | 69 (37.10) | 56 (35.22) | 13 (48.15) | 0.204 |
| Radiotherapy | 21 (11.29) | 20 (12.58) | 1 (3.70) | 0.320 |
| Targeted therapy | 6 (3.23) | 4 (2.52) | 2 (7.41) | 0.210 |
| Immunotherapy | 11 (5.91) | 10 (6.29) | 1 (3.70) | 1.000 |
| Severity of COVID-19 infection |  |  |  |  |
| Mild | 134 (72.04) | 125 (78.62) | 9 (33.33) | <0.00001 |
| Moderate | 43 (23.12) | 32 (20.13) | 11 (40.74) |  |
| Severe | 9 (4.84) | 2 (1.26) | 7 (25.93) |  |
| Patients who received ventilator support | 12 (6.45) | 0 (0.00) | 12 (44.44) | 0.000 |

significantly higher in patients presented with severe COVID-19 infection symptoms compared to mild/moderately symptomatic patients (OR = 27.48, $P < 0.001$). Patients who were on active cancer treatment during 1 month before contracting COVID-19 did not have an increased risk of death ($P = 0.460$). Also, the treatment modalities including surgery, chemotherapy, radiotherapy, targeted therapy, and immunotherapy did not confer an increased risk of death in univariate analyses. There was a significant difference in the median age of the patients who received chemotherapy compared to those who did not (49 years vs. 54 years, $P = 0.005$). So we further examined whether anticancer therapies could influence mortality in the cohort by adjusting for age and comorbidity in the multivariate logistic regression analysis (Table 3). In comparison to the patients who were not on these treatments, there were no significant increase in risk of death with chemotherapy (OR = 1.63, 95% CI [0.64–4.15], $P = 0.301$), radiotherapy (OR = 0.19, 95% CI [0.02–1.59], $P = 0.126$), targeted therapy (OR = 2.70, 95% CI [0.42–17.37], $P = 0.296$), immunotherapy (OR = 0.40, 95% CI [0.04–3.56], $P = 0.413$), or surgery (OR = 0.87, 95% CI [0.25–2.94], $P = 0.819$) (Table 3).

# DISCUSSION

The COVID-19 pandemic has raised several fears. One of these has been the increased risk of cancer patients to contract COVID-19. In the current study, we found the incidence of COVID-19 to be ~6.0% in our hospitalized cancer patients. The national incidence of COVID-19 in an unselected cohort is 0.32% and 0.29% as per the European Centre for Disease Prevention and Control (Fig. 1A) (*Government of India, 2020*; *Roser et al., 2020*). The incidence of COVID-19 observed in hospitalized cancer patients is expected to be higher than in the general population. This on the one hand can be ascribed to patients' factors, visitations, and admission to a health care facility with a high risk of contracting infection; may also be partly because of lower case detection rates in the general population

**Table 2 Logistic regression analysis (univariate) and odds ratio for death in the cohort (N = 186 COVID-19 infected cancer patients).**

| Variable | Odds ratio | 95% CI | P value |
|---|---|---|---|
| Age | 1.00 | [0.97–1.03] | 0.974 |
| Gender (male vs. female) | 1.14 | [0.49–2.62] | 0.750 |
| Comorbidity | 2.68 | [1.13–6.32] | 0.025 |
| Comorbidity (multiple vs. single) | 3.01 | [1.02–9.07] | 0.047 |
| Cardiovascular disease | 3.77 | [1.02–13.91] | 0.046 |
| Diabetes | 3.31 | [1.35–8.09] | 0.009 |
| Hypertension | 1.71 | [0.71–4.13] | 0.234 |
| Thyroid (hypo/hyper) | 0.89 | [0.28–2.79] | 0.841 |
| Other comorbidities | 4.16 | [0.66–26.15] | 0.129 |
| Hematological malignancies vs. solid tumors | 1.79 | [0.69–4.67] | 0.233 |
| Head and neck | 1.40 | [0.51–3.79] | 0.511 |
| Breast | 0.30 | [0.04–2.36] | 0.253 |
| Musculoskeletal and skin | 3.10 | [0.54–17.82] | 0.205 |
| Gastrointestinal | 2.06 | [0.85–5.03] | 0.111 |
| Genitourinary and gynecologic | 0.29 | [0.07–1.30] | 0.107 |
| Advanced vs. localized cancer | 1.42 | [0.50–4.10] | 0.514 |
| Metastatic vs. localized cancer | 2.17 | [0.73–6.46] | 0.166 |
| Cancer treatment | 1.38 | [0.58–3.27] | 0.460 |
| Surgery | 0.85 | [0.27–2.66] | 0.780 |
| Chemotherapy | 1.71 | [0.75–3.88] | 0.202 |
| Radiotherapy | 0.27 | [0.03–2.08] | 0.208 |
| Targeted therapy | 3.10 | [0.54–17.82] | 0.205 |
| Immunotherapy | 0.57 | [0.07–4.67] | 0.603 |
| Mild COVID-19 infection | 0.14 | [0.06–0.33] | <0.0001 |
| Moderate COVID-19 infection | 2.72 | [1.15–6.45] | 0.022 |
| Severe COVID-19 infection | 27.48 | [5.34–141.49] | <0.001 |

Note:
   For the categorical variables where the comparison group is not indicated absence was taken as the reference.

**Table 3 Multivariate logistic regression analysis and risk of death in the cohort (N = 186 COVID-19 infected cancer patients).**

| Variable | Odds ratio | 95% CI | P value |
|---|---|---|---|
| Age | 0.99 | [0.96–1.02] | 0.403 |
| Comorbidity | 3.16 | [1.16–8.61] | 0.024 |
| Chemotherapy, Yes/No | 1.63 | [0.64–4.15] | 0.301 |
| Radiotherapy, Yes/No | 0.19 | [0.02–1.59] | 0.126 |
| Targeted therapy, Yes/No | 2.70 | [0.42–17.37] | 0.296 |
| Immunotherapy, Yes/No | 0.40 | [0.04–3.56] | 0.413 |
| Surgery, Yes/No | 0.87 | [0.25–2.94] | 0.819 |

**A**

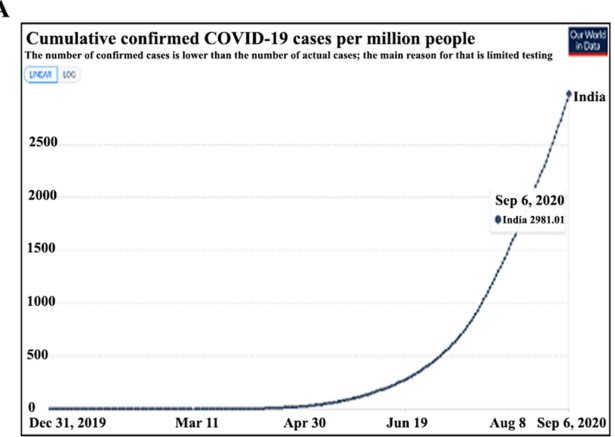

**B**

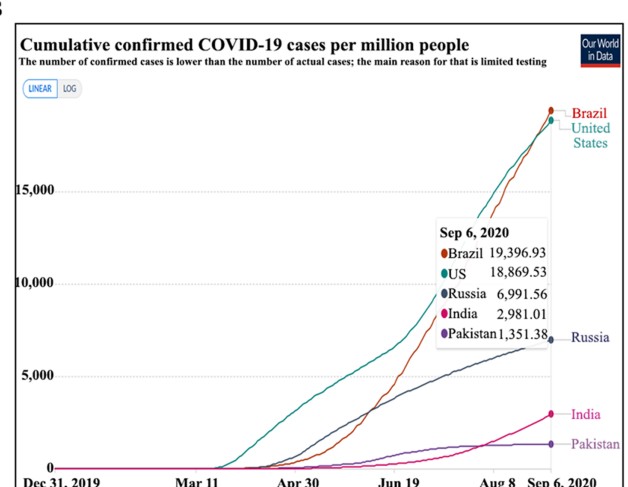

**C**

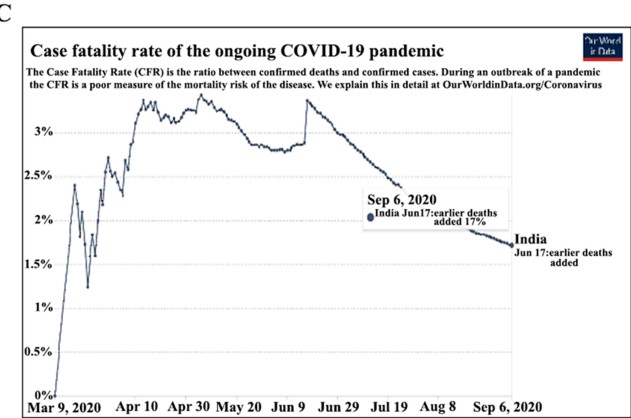

**Figure 1** **Incidence and fatality due to COVID-19 in the population (A) cumulative COVID-19 incidence per million population (India). (B) The incidence of COVID 19 in general population ranges from 1.9% in Brazil to 0.13% in Pakistan. (C) The COVID-19 Case Fatality Rate in India is 1.7%.** Source: European Centre for Disease Prevention and Control (https://www.ecdc.europa.eu/en/publications-data/download-todays-data-geographic-distribution-covid-19-cases-worldwide. Data as on September 6, 2020) (*Roser et al., 2020*).

in India due to poor access to health care facilities and low testing rates. The reported incidence in other high COVID-19 incidence nations like Brazil and the USA, and other middle and low incidence countries have been shown in Fig. 1B.

The average CFR of COVID-19 is 1.9% in the unselected Indian population as per the national database (Arogyasetu Application), 1.84% by Johns Hopkins COVID-19 tracker, and 1.7% as per the European Centre for Disease Prevention and Control (Fig. 1C) (*Government of India, 2020*; *John Hopkins University and Medicine Coronavirus Resource Center, 2020*; *Roser et al., 2020*). The 14.52% COVID-19 related CFR observed for hospitalized cancer patients in the present study is similar to the data reported by *Ma et al. (2020)* from Renmin Hospital of Wuhan University (CFR-13.5%), and is higher than a recent report from Tata Memorial Hospital, Mumbai, India (CFR-6.5%) (*Ma et al., 2020*; *Ramaswamy et al., 2020*). The previous studies from the European continent have also shown a far greater CFR in cancer patients with COVID-19. In the UK Coronavirus Cancer Monitoring Project (UKCCMP), a CFR of 30.6% was observed, where 319 of the 1,044 cancer patients with COVID-19 died with 92.5% had their death attributed directly to COVID-19 (*Lee et al., 2020*). In another observational study by *Pinato et al. (2020)* of 890 cancer patients diagnosed with SARS-CoV-2, mortality was found to be 33.6%. Similarly, high rates were noticed in a New York Hospital System where a CFR of 28% was observed (*Mehta et al., 2020*). A large systematic review of 52 studies by a group led by Kamal S. Saini involving more than 18,000 cases of cancer with COVID-19, the probability of death was 25.6% (95% CI [22.0–29.9]; $I^2$ = 48.9%) (*Saini et al., 2020*). Among the hospitalized cancer patients, we observed that the CFR was about 10.6 times higher in the COVID-19 infected patients than those without COVID-19 (14.52% vs. 1.37%, $P < 0.0001$) indicating that COVID-19 infection significantly increased the risk of death in the cohort.

Unlike the previous retrospective studies from China, we did not find any significant association between recent cancer treatments and mortality (*Dai et al., 2020*; *Liang et al., 2020*). Our observation is in line with two large cohort studies conducted by *Lee et al. (2020)* on 800 COVID-19 positive cancer patients from UKCCMP, and by *Robilotti et al. (2020)* on 423 symptomatic COVID-19 cancer patients at Memorial Sloan Kettering Cancer Centre in New York (*Lee et al., 2020*; *Robilotti et al., 2020*). The recent multi-center study by *Pinato et al. (2020)* in the European cancer patients and the Covid-19 and Cancer Consortium (CCC19) database study also strengthens the notion that cancer treatment is not associated with mortality (*Kuderer et al., 2020*; *Pinato et al., 2020*). Similar to these studies we found that it is the underlying comorbid conditions and COVID-19 disease severity that are associated with adverse outcomes. Consistent with the CCC19 study we observed a higher burden of pre-existing comorbidities (>1) to be associated with increased mortality (*Kuderer et al., 2020*). Cardiovascular diseases and diabetes posed a higher risk of death in our cohort. It should be noted that comorbid conditions like diabetes frequently co-occur with hypertension or coronary artery disease in patients and can further weaken the immune response escalating the risk of death due to COVID-19 (*Guan et al., 2020*; *Naqvi et al., 2017*).

## CONCLUSION

Our study highlights the high rates of COVID-19 in cancer patients, with a CFR of 14.52%. Recent anticancer therapies did not have a significant effect on mortality in the cohort. Pre-existing comorbidities especially diabetes, presence of more than one comorbidity and severe COVID-19 presenting symptoms were significantly linked with COVID-19 related deaths in the cohort.

### Funding
The authors received no funding for this work.

### Competing Interests
The authors declare that they have no competing interests.

### Author Contributions
- Anurag Mehta conceived and designed the experiments, prepared figures and/or tables, authored or reviewed drafts of the paper, and approved the final draft.
- Smreti Vasudevan conceived and designed the experiments, analyzed the data, prepared figures and/or tables, authored or reviewed drafts of the paper, and approved the final draft.
- Anuj Parkash conceived and designed the experiments, authored or reviewed drafts of the paper, and approved the final draft.
- Anurag Sharma analyzed the data, authored or reviewed drafts of the paper, statistical analysis, and approved the final draft.
- Tanu Vashist performed the experiments, prepared figures and/or tables, and approved the final draft.
- Vidya Krishna performed the experiments, prepared figures and/or tables, and approved the final draft.

### Human Ethics
The following information was supplied relating to ethical approvals (i.e., approving body and any reference numbers):

Institutional Review Board, Rajiv Gandhi Cancer Institute and Research Centre, India approved the study (RGCIRC/IRB-BHR/61/2020).

### Data Availability
The datasets are available in a Supplemental File.

### Supplemental Information
Supplemental information for this article can be found online at http://dx.doi.org/10.7717/peerj.10599#supplemental-information.

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
