# Peer review of "COVID-19 mortality in cancer patients: a report from a tertiary cancer centre in India"

_PeerJ, doi:10.7717/peerj.10599_

## Round 0.1 · original submission · Major Revisions

The reviewers have raised interest in this work and have found the study well-conducted and of interest to the community. However, they have also raised some concerns about the interpretation and reporting of the results, which should be addressed before the manuscript can be considered for publication.

Reviewer 1 ·

Basic reporting

I think it is a well-designed and well-written paper. The aims and design are clear. The discussion is well-structured with up-to-date references. However, I think that some comments should be added about the collateral damage in the introduction and discussion and about the mechanism of increased COVID-19 contracting risk in cancer patients as we did in one of our papers (Guven DC, Aktas BY, Aksun MS, et al. COVID-19 pandemic: changes in cancer admissions. BMJ Supportive & Palliative Care. Published Online First: 14 July 2020. doi: 10.1136/bmjspcare-2020-002468). The English is generally very good but I added some suggestions and highlighted them via the track changes feature in order to further improve some sentences. It's up to the authors to use or not use these changes. The figures and tables are largely acceptable but the reporting of values should be uniform.

Experimental design

I think the study is in the scope of the journal. The study tries to answer a clinically meaningful and important question. The authors tested a significant number of patients before study inclusion. The patient numbers are adequate for conducting the analyses. The methodology is explained in detail.

Validity of the findings

Although the study hypothesis was not a novel hypothesis, it is a very important study considering the relative paucity of data from the region of the study. This study could add to the literature about the situation of the pandemic in low-resource settings like India. Conclusions are well written and are generally in-line with previous research in similar patient cohorts.

Additional comments

First of all, I want to congratulate the authors for conducting this elegant and precious work while dealing with the pandemic in a country which was significantly hit with the pandemic. I think the study is a well-constructed and well-designed study with enough. However, it could be further improved with minor changes. I think that some comments should be added about the collateral damage in the introduction and discussion and about the mechanism of increased COVID-19 contracting risk in cancer patients as we did in one of our papers. The English is generally very good but I added some suggestions and highlighted them via the track changes feature in order to further improve some sentences. It's up to the authors to use or not use these changes.

Annotated reviews are not available for download in order to protect the identity of reviewers who chose to remain anonymous.

·

Basic reporting

No comment

Experimental design

No comment

Validity of the findings

Overall, your manuscript is informative, but some edits/improvements need to be made to correctly convey the data:

• Discussion:
The 6% incidence in COVID-19 in your study is based on HOSPITALIZED cancer patients and not cancer patients, in general. Therefore, your comparison with the general populations of India and Europe is relatively meaningless since one would expect a higher incidence of COVID-19 for all hospitalized patients, whether or not they have cancer. Comparison of COVID-19 incidence in your institution’s outpatient department to the national incidence would likely be more appropriate and would better quantify the effect of the presence of cancer, itself, on the higher risk of COVID-19 infection.

The same point can be made for your stated CFR of 14.5% vs the national average of 1.9% -- you are comparing hospitalized patients to non-hospitalized + hospitalized patients. The comparison gives an incorrect impression of a very pronounced difference. If this is not the case, then please better specify the patient populations being compared.

• Conclusion:
Your conclusions reflect the exaggeration of COVID-19 incidences that I just discussed and should be appropriately modified. Comparison of your data with NON-CANCER patients hospitalized in India and elsewhere would better tell the story of how the presence of cancer impacts COVID-19 incidence.

• Results:
Providing a comparison of the fatality rates for hospitalized cancer patients with and without COVID-19 would also provide a level field for comparison of the effect of COVID-19 on fatality on hospitalized cancer patients.

Was the difference in CFRs for those with solid vs hematological cancers statistically significant? If so, this is an important finding and worthy of more investigation (not necessarily in this report) to discover why this is the case. If it is not, I suggest deleting the sentence since it gives the false impression of a true difference – or provide a p-value for this statement.

Additional comments

No comment

---

## Round 0.2 · accepted · Accept

In preparing your manuscript for publication, I highly recommend changing the figures so they are not screenshots and are the reproductions of the data obtained from public resources. Congratulations on acceptance of your article.

Reviewer 1 ·

Basic reporting

I think the paper is significantly improved and acceptable after this revision.

Experimental design

I think the paper is significantly improved and acceptable after this revision.

Validity of the findings

I think the paper is significantly improved and acceptable after this revision.

Additional comments

I think the paper is significantly improved and acceptable after this revision.